# OpenReview forum: "Corrupted by Reasoning: Reasoning Language Models Become Free-Riders in Public Goods Games"
_colmweb.org/COLM/2025/Conference — COLM 2025_

### Official Review · Reviewer_gBFf · 2025-05-04

**Rating:** 7
**Confidence:** 4
**Ethics Flag:** 1

**Summary:**

This paper investigates how LLMs behave in multi-agent social dilemmas, specifically focusing on costly sanctioning in public goods games with institutional choice. The study evaluates whether LLM agents are willing to invest resources to punish defectors or reward cooperators, a key mechanism humans use to sustain cooperation. Through extensive experiments on a range of LLM families (traditional vs. reasoning-enhanced models), the authors reveal surprising findings: ``Traditional LLMs often maintain high cooperation levels using sanctioning institutions.’’

Reasoning-enhanced LLMs (e.g., o1 series) more often collapse into non-cooperation, prioritizing individual rationality over group welfare. The paper identifies four behavioral archetypes among the models and offers qualitative insights into the underlying reasoning patterns that drive cooperative or selfish behaviors.

**Questions To Authors:**

- How robust are the results to different phrasings of the public goods and sanctioning tasks? How to guarantee that LLM agent totally follow the prompt? Would slightly different language alter cooperation rates?
- Have you attempted to analyze the hidden representations or intermediate activations of cooperative vs. defecting models to understand how the failure modes emerge internally? What are the feasible scheme and potential outcomes?
- Based on your findings, what directions seem most promising for designing LLMs that maintain cooperation? Could architectural changes, fine-tuning objectives, or scaffolded prompting help?
- Do you plan to test the models in other cooperation scenarios, like trust games, negotiation games, or distributed control settings? Would the same failure modes emerge?

**Reasons To Accept:**

- This paper explores a timely and important problem. Understanding cooperation in LLMs is critical for safe, scalable deployment in multi-agent systems and societal settings.
- The use of public goods games with institutional choice adapted to LLMs is clever and connects AI evaluation with behavioral economics.
- This paper challenges common assumptions that better reasoning always leads to better social outcomes, which is a crucial insight for LLM alignment research.
- This paper provides a thorough analysis across multiple models and identifies distinct behavioral archetypes, rather than focusing narrowly on a single trend.
- This paper highlights real risks for governance and collaboration among AI agents, suggesting paths for future intervention research.

**Reasons To Reject:**

- The paper doesn’t clearly specify how sensitive outcomes are to the way prompts and agent communication are designed, prompt phrasing could critically bias model behavior.
- While behavioral outcomes are categorized, the internal model states, representations, or learning dynamics that lead to these behaviors are not probed deeply (e.g., hidden states, attention patterns).
- Although human cooperation patterns are mentioned, the study does not provide side-by-side comparisons to real human experiments under the same setup. In the Ethics Statement, the authors mentioned that `` This study involves only simulations with LLM-based agents in public goods games and does not include human participants.’’

---

> ### Author Response · Authors · 2025-06-02
> **Response to Reviewer gBFf**
>
> Thank you for your strong support and insightful comments. We have simplified your comments for easier reference and included our respective responses. We will carefully address your concerns one by one.
>
> *   **Prompt Sensitivity/Robustness & LLM Adherence:**
>
>     As mentioned in our response to R-cqML, our prompts were based on human studies for comparability. We conducted **preliminary tests with an alternative prompt framing**, and initial results suggest our core findings on behavioral archetypes are robust.
>
>     The structured JSON output and consistent reasoning patterns within archetypes (especially for models run multiple times) suggest they generally comprehend and engage with the task as intended.
>
> *   **Internal Model State Analysis:**
>
>     Thank you, this is an excellent suggestion. Analyzing internal model states like hidden representations to understand how these behaviors emerge is indeed a significant research direction. Our current paper focuses on characterizing the emergent behavioral dynamics, we agree this interpretability work is a valuable avenue for future investigation.
>
> *   **Human Comparison:**
>
>     To clarify, while we did not run new human experiments (per our ethics statement and scope), a **key strength of our study is the direct comparison of LLM agent behavior with well-established human data** from Gurerk et al. [1] and large-scale replications (e.g., Iacono et al. [2]). Our PGG setup was designed to be highly similar, allowing for benchmarking against documented human tendencies in these specific social dilemmas.
>
> *   **Designing Cooperative LLMs:**
>
>     As mentioned in the General Response and our response to R-WHGr, promising directions include pro-social fine-tuning, reinforcement learning with group welfare objectives, and exploring Multi-Agent architectural changes for collective reasoning.
>
> *   **Testing in Other Cooperation Scenarios:**
>
>     As mentioned in the General Response, we are keen to extend this methodology to other games like trust games, negotiation, or common dilemmas to see if similar behavioral patterns and failure modes emerge.
>
> **References**
>
> [1] Gurerk, O., Irlenbusch, B., & Rockenbach, B. (2006). The Competitive Advantage of Sanctioning Institutions. *Science*.
>
> [2] Iacono, D., Ross, L., & Ziegelmeyer, A. (2023). Replication of Gürerk, Irlenbusch, and Rockenbach (2006): The competitive advantage of sanctioning institutions. *PNAS Nexus*.

---

> > ### Comment · Reviewer_gBFf · 2025-06-03
> > **Thank You for the Response**
> >
> > Dear Authors,
> >
> > Thank you very much for the clarification! Most of them have addressed my concerns.
> >
> > I appreciate the response, and highlight those explanation in the revised draft will strengthen the paper.
> >
> > Overall, I acknowledge the interesting idea of this work, but still have reservations about experiments on **Human Comparison** (How do you guarantee that real human experiments are under the same setup with agent experiments? Just comparing with previous human experiment data isn't convincing and sound enough), and decide to keep my positive score.
> >
> > Best Regards,
> >
> > Reviewer gBFf

---

> > > ### Author Response · Authors · 2025-06-03
> > >
> > > Thank you for your positive assessment and for acknowledging our responses. We appreciate your feedback, particularly regarding the human comparison. To address your point about the setup, we will emphasize in the revised manuscript that the human experiments (e.g., Gurerk et al. [1]) also featured computer-based interactions with textual instructions and a well-defined, finite set of available actions. Humans were placed within individual booths, precluding any verbal or non-verbal communication between participants. This creates an action space and information environment very similar to that of our LLM agents, which operate based on textual instruct instructions, have access to the same set of action choices, and also lack direct inter-agent communication. We will also more clearly acknowledge the natural differences in cognitive architecture (e.g., human biology vs transformers, human social norms vs heuristics learned in pre-training, etc). Your suggestions have been very helpful in refining this aspect!

---

### Official Review · Reviewer_cqML · 2025-05-06

**Rating:** 7
**Confidence:** 3
**Ethics Flag:** 1

**Summary:**

The paper studies the cooperation of LLM-driven agents in social dilemmas by simulating public good games and analyzing the results across multiple conditions. The experimental setup and analysis reveal that reasoning LLMs cooperate less compared to traditional LLMs. The paper also compares the behavior of LLMs to that of human players and analyzes how and when cooperation succeeds or fails over time.

**Questions To Authors:**

1. Is it really surprising that reasoning LLMs achieve lower cooperation rates compared to traditional LLMs? If the goal of the model is to optimize its own utility, we expect reasoning models to behave relatively close to the game-theoretic agents, right? I believe that this observation is important, even if not surprising, but I want to make sure I am not missing anything here.

2. It is mentioned that agents did not engage in any natural language communication through the interaction. Is it always the case in real-world applications? Would you expect that some of the results will change if we allow for language-based communication?

UPDATED: after carefully reviewing the authors' response, I have decided to raise my score from 6 to 7.

**Reasons To Accept:**

1. The paper studies a timely and important topic, and builds on a well-established framework.

2. The paper is well written and very easy to follow.

3. The experimental design is rigorous, and the accompanying analysis appears robust and well-executed.

**Reasons To Reject:**

1. The scope of the paper is limited to the specific framework of public good games. In practice, cooperation potentially appears in a more diverse range of economic games (for instance, repeated prisoner’s dilemma). This naturally means that the conclusions drawn by this work may not extend beyond the scope of this specific game. I do not consider this a major limitation, as the scope of a research paper is limited by nature, but I still consider it as a weakness. Perhaps it is worth at least discussing potential forms of cooperation in other economic contexts (preferably ones in which using LLMs is natural) and describing how the methodology presented in this work could be applied to study LLMs cooperation.

2. Even if we agree on the importance and relevance of the limited scope of public good games, it appears that the prompting approach is somewhat synthetic. The prompts are presented in purely technical terms, without grounding the game in real-world scenarios or intuitive narratives. This abstraction may reduce validity and raise concerns about the robustness of the results. For example, rephrasing the same public good dilemma using different contextual framings (e.g., climate action, shared resource funding) could potentially affect the LLMs’ behavior. These aspects might be crucial when considering the cooperation of LLM-driven agents in real-world systems.

---

> ### Author Response · Authors · 2025-06-02
> **Response to Reviewer cqML**
>
> Thank you for your valuable feedback and for recognizing the paper's clarity and design. We have simplified your comments for easier reference and included our respective responses. We will carefully address your concerns one by one.
>
> *   **Limited Scope (Public Goods Games):**
>
>     Thanks for the feedback! As mentioned in the Global Response, our choice of the Public Goods Game (PGG) was deliberate. It is an extensively studied paradigm, and using it allows for direct comparison with robust human behavioral data from Gurerk et al. [1] and subsequent large-scale replications (e.g., Iacono et al. [2]). We agree that extending this methodology to other economic games, such as the repeated prisoner's dilemma or trust games, is an important avenue for future work to explore the generality of these behavioral patterns.
>
> *   **Synthetic Prompting:**
>     Our prompts were designed to be **technical and directly adapted from the verbatim instructions given to human participants** in Gurerk et al. [1]. This was a choice to ensure a controlled environment, minimize confounding variables from narrative framing, and facilitate direct human-LLM comparison.
>
>     To address prompt sensitivity, we ran **tests with an alternative prompt** incorporating a narrative framing for key models (Llama-3.3-70B and o1-mini). This narrative framing reframed the game as a **'community project initiative' where agents were 'community members' making 'investments.'** Institutions were described as 'Independent' vs. 'Accountable' groups, and Stage 2 sanctioning involved assigning 'punishment tokens' or 'reward tokens'.
>
>     Our findings from this narrative prompt test suggest that the **core behavioral patterns largely persisted** for both Llama-3.3-70B (maintaining high cooperation) and o1-mini. This indicates that our main conclusions are not solely an artifact of the initial technical phrasing.
>
>     Further details and a visual comparison are available in the figure and table below.
>
>     **Figure 2: Impact of Narrative Prompting on LLM Contribution Trajectories**
>     (Please see: https://imgur.com/a/F4uUMnt)
>
>     | Condition             | Llama 3.1 70B Avg. Contr.  | o1-mini Avg. Contr. | Llama 3.1 70B  SI % | o1-mini SI % | Llama 3.1 70B % Free Riders | o1-mini % Free Riders |
>     | :-------------------- | :----------------------------------------: | :-----------------------------------: | :--------------------: | :--------------: | :-----------------------------: | :------------------------: |
>     | **Baseline (Original)** | **18.71 / 20 (93.5%)**                     | **5.39 / 20 (26.9%)**                 | **99.6**               | **28.0**         | **0.0**                         | **69.3**                   |
>     | Narrative Prompt      | 18.79 / 20 (94.0%)                         | 9.95 / 20 (49.8%)                     | 100.0                  | 28.6             | 0.0                             | 45.7                       |
>
>     *Table 2: Impact of Narrative Prompting on Key Cooperation Metrics vs. Baseline. Baseline values are from the original study. Narrative prompt results are from a single simulation run.*
>
> *   **Surprisingness of Reasoning LLM Low Cooperation:**
>
>     Thanks! This is a key observation. It's noteworthy that:
>
>     *   While game theory predicts self-interested behavior for rational agents, the key is that this "better" reasoning, can lead to collectively worse outcomes, **mimicking a narrower form of mathematical rationality rather than more human-like cooperative intelligence**. This is particularly interesting given the prompts never mentioned any clear objectives (pro-social or not).
>
>     *   Furthermore LLMs were **not explicitly instructed to optimize individual utility**; this  behavior emerged naturally, particularly strongly in reasoning models.
>
>     *   This **challenges the assumption** that enhanced "reasoning" capabilities may directly translate to beneficial social behavior. Instead, it can lead to outcomes that are collectively worse, highlighting a critical alignment consideration.
>
> *   **Natural Language Communication:**
>
>    We acknowledge the importance of NL communication in many real-world scenarios. Its omission in our current study was to first establish **baseline behaviors in a controlled setting with fewer confounders** as well as adhering to design choices from the human experiments (Gurerk et al. [1]). Introducing communication is a rich area for future work and could significantly impact dynamics, potentially fostering cooperation or enabling more sophisticated defection.
>
> **References**
>
> [1] Gurerk, O., Irlenbusch, B., & Rockenbach, B. (2006). The Competitive Advantage of Sanctioning Institutions. *Science*.
>
> [2] Iacono, D., Ross, L., & Ziegelmeyer, A. (2023).

---

> > ### Comment · Reviewer_cqML · 2025-06-04
> > **Response to reviewers' reply**
> >
> > Thank you for the thoughtful response to my comments and concerns. I have raised my score from 6 to 7, and I suggest incorporating these points into the next version of the paper.

---

> > > ### Author Response · Authors · 2025-06-04
> > >
> > > Dear Reviewer cqML,
> > >
> > > We're delighted to hear that most of your concerns have been addressed. Thank you again for the time and effort you dedicated to reviewing our paper. We'll incorporate your feedback in the revision. If you have any further questions, please don't hesitate to ask.
> > >
> > > Best regards,
> > >
> > > The Authors

---

### Official Review · Reviewer_WHGr · 2025-05-11

**Rating:** 7
**Confidence:** 4
**Ethics Flag:** 1

**Summary:**

This paper investigates the cooperative behaviors of multi-agent LLMs. It develops a public goods game where agents must decide whether to invest their own resources to incentivize cooperation or penalize defection. The study reveals four distinct behavioral patterns among different LLMs: consistent high cooperation, fluctuating engagement, declining cooperation, and rigid adherence to fixed strategies. A key finding is that reasoning models surprisingly struggle with cooperation, while some traditional LLMs achieve high levels of cooperation, suggesting that enhanced reasoning capabilities do not inherently lead to better cooperative outcomes.

**Questions To Authors:**

Minor typo:
Line 210: not -> now

1. Following weakness, how sensitive do you anticipate your findings to be regarding the specific parameterization of the public goods game (e.g., payoff structures, cost/impact of sanctions)?
2. Given the finding that enhanced reasoning does not necessarily lead to cooperation, what specific research directions or alignment strategies do you foresee as most promising for developing LLM agents that can reliably cooperate and contribute to the collective good in multi-agent systems?

**Reasons To Accept:**

1. The paper addresses an increasingly important and timely research question: how LLMs navigate social dilemmas and the balance between self-interest and collective well-being. This is critical for the safe and reliable deployment of LLM-based systems.
2. The experiment setup is well-grounded. It adapts a recognized experimental paradigm from behavioral economics to the LLM context. This allows for a structured study of complex cooperative dynamics.
3. I like how the paper provides a comparative analysis with human behavior in similar public goods games. This comparison is particularly insightful as it highlights both convergent outcomes, such as achieving high cooperation levels and a strong tendency to migrate towards sanctioning institutions, which mirrors human behavior in these scenarios. However, the paper also points out divergent strategies, most notably the LLMs' distinct preference for using rewards to foster cooperation, which contrasts sharply with the human tendency to rely more on punishment. This juxtaposition of similarities and differences in approach offers valuable perspectives on the emerging nature of LLM "sociality" and how these artificial agents navigate complex social interactions differently from humans, even when achieving similar overall results.

**Reasons To Reject:**

1. The study uses a specific set of game parameters (e.g., endowment size, multiplication factor, sanctioning costs/effects). While standard, a discussion on the sensitivity of the findings to variations in these parameters is required. Would reasoning LLMs perform differently if the "price" of cooperation or the "cost" of sanctioning were altered significantly?

---

> ### Author Response · Authors · 2025-06-02
> **Response to Reviewer WHGr**
>
> Thank you for your assessment and constructive feedback. We have simplified your comments for easier reference and included our respective responses. We will carefully address your concerns one by one.
>
> *   **Parameter Sensitivity & Robustness:**
>     We agree this is an important consideration. Our experimental parameters were adopted from the established Gurerk et al. [1] study to ensure direct comparability with human behavioral data.
>
>     As introduced in the General Response, rebuttal tests altering key game parameters confirmed that the **fundamental behavioral archetypes of Llama-3.3-70B (sustained cooperation) and o1-mini (defection) remained consistent**, with Llama-3.3-70B generally maintaining high contributions and o1-mini continuing its defecting strategy. The following figure illustrates how Llama-3.3-70B converges to consistent contributions while o1-mini converges to widespread defection, with varied parameters.
>
>     **Figure 1: Impact of Parameter Variations on LLM Cooperation Trajectories**
>     (Please see: https://imgur.com/a/xINp746)
>
>     A comprehensive sensitivity analysis is valuable future work, and we will include these initial findings in the revised manuscript's appendix. More details on our ablation are contained in the following table:
>
> | Condition                 | Avg. Contr. (Tokens & %) |                           | SI %                |                 | % Free Riders             |                           |
> | :------------------------ | :---------------------------: | :--------------------: | :------------------: | :--------------: | :------------------------: | :--------------------: |
> |                           | Llama 3.1 70B            | o1-mini                   | Llama 3.1 70B       | o1-mini         | Llama 3.1 70B             | o1-mini                   |
> | **Baseline (Original)** | **18.71 / 20 (93.5%)** | **5.39 / 20 (26.9%)** | **99.6** | **28.0** | **0.0** | **69.3** |
> | Multiplier: Low (α=1.2)   | 18.73 / 20 (93.7%)        | 3.95 / 20 (19.8%)         | 99.0                 | 26.7             | 0.0                      | 76.2                      |
> | Multiplier: High (α=2.5)  | 18.80 / 20 (94.0%)        | **1.43 / 20 (7.1%)** | 100.0                | **12.4** | 0.0                      | **90.5** |
> | Punishment: Weak (C1,E-1) | 18.80 / 20 (94.0%)        | 6.14 / 20 (30.7%)         | 100.0                | 35.2             | 0.0                      | 66.7                      |
> | Punishment: Costly (C3,E-3)| 18.47 / 20 (92.3%)       | 3.33 / 20 (16.7%)         | 100.0                | 20.0             | 0.0                      | 81.9                      |
> | Endowment: Low (e=10)     | 9.47 / 10 (94.7%)         | 4.96 / 10 (49.6%)         | 100.0                | 50.5             | 6.7                      | 51.4                      |
> | Endowment: High (e=40)    | 34.33 / 40 (85.8%)        | 6.05 / 40 (15.1%)         | 100.0                | 19.0             | 0.0                      | 81.9                      |
>
>     *Table 1: Preliminary Ablation Study Results - Impact of Parameter Variations on Key Cooperation Metrics. Baseline values are from the original study. Ablation results are from single simulation runs. "Avg. Contr." shows average tokens contributed out of the maximum possible for that endowment, with the percentage of maximum in parentheses. "SI %" is the percentage of rounds agents chose the Sanctioning Institution. "% Free Riders" is the percentage of agent-rounds contributing 5 or fewer tokens.*
>
> *   **Future Directions for Cooperative LLMs:**
>
>     Based on our findings, and expanding further on the points mentioned in the General Response, promising directions include:
>
>     *   Pro-social fine-tuning: Explicitly training or fine-tuning LLMs on datasets or with objectives that reward collective well-being.
>
>     *   Exploring communication: Investigating how structured communication might alter cooperative dynamics.
>
>     *   Architectural considerations: Designing multi-agent scaffolding with mechanisms that heavily weigh group outcomes.
>
> *   **Typo (Line 210):**
>     Thank you for pointing that out! We’ll correct it.
>
> **References**
>
> [1] Gurerk, O., Irlenbusch, B., & Rockenbach, B. (2006). The Competitive Advantage of Sanctioning Institutions. *Science*.

---

> > ### Comment · Reviewer_WHGr · 2025-06-11
> > **Rebuttal Response**
> >
> > I thank the authors for the detailed and thoughtful response. Adding more ablations enhances the validity of the experiment conclusions. Overall I believe it is a good paper ready to publish. I enourage the authors to include the results in the next version.

---

### Author Response · Authors · 2025-06-02
**General Response**

We thank all reviewers for their time and feedback. We are grateful that the reviewers recognized the **timeliness and importance of our work** in understanding LLM cooperation (R-WHGr, R-cqML, R-gBFf), the **rigorous experimental design** and **well-grounded setup** (R-WHGr, R-cqML), the **comparison with human behavior**, particularly the divergent reward/punishment strategies (R-WHGr), and that our findings **challenge common assumptions** about reasoning leading to better social outcomes (R-gBFf).

Our core contributions are the identification of four distinct behavioral patterns in LLM cooperation and the critical finding that current **reasoning LLMs can be less effective at sustaining cooperation** than traditional models. These reasoning models often prioritize individual rationality in a way that undermines collective good, suggesting their enhanced analytical capabilities do not inherently lead to **human-like cooperative intelligence or social reasoning**, which often navigates such dilemmas more successfully. This highlights an important consideration for LLM alignment.

Several reviewers raised excellent and shared points, which we address here globally and in more detail in individual responses:

*   **Sensitivity to Experimental Setup (Game Parameters & Prompt Phrasing) (R-WHGr, R-cqML, R-gBFf):**
    *   Our primary experimental parameters and prompt phrasing were adopted from the established Gurerk et al. [1] human study. This ensures direct comparability with well-documented human behavioral data and provides a controlled environment to isolate core decision-making.

    *   During this rebuttal period, we conducted **targeted experiments** altering key game parameters (public good multiplier, Punishment Cost & Effect, Endowment) and testing an alternative prompt with a narrative framing on representative models (Llama-3.3-70B and o1-mini). This narrative framing reframed the game as a 'community project initiative' where agents were 'community members' making 'investments.' Institutions were described as 'Independent' vs. 'Accountable' groups, and Stage 2 sanctioning involved assigning 'punishment tokens' or 'reward tokens' (retaining these traditional terms for Stage 2 actions as per our refined ablation).

    *   Our initial findings from these new tests suggest that the **fundamental behavioral archetypes – sustained cooperation for Llama-3.3-70B and defection for o1-mini – remained largely consistent**, indicating a degree of robustness to these specific variations. We will include these details and the full narrative prompt in the appendix of the revised manuscript. Further details on parameter variations can be found in the individual responses and the summary table provided to Reviewer WHGr.

    *   We agree that a comprehensive sensitivity analysis across a wider range of parameters and prompt styles is valuable future work.

*   **Scope of Study (Public Goods Games) & Future Scenarios (R-cqML, R-gBFf):**

    *   We chose Public Goods Games (PGGs) as they are a **foundational and extensively studied paradigm** for social dilemmas (Gurerk et al. [1]), allowing for robust benchmarking against human behavior.

    *   We agree that extending this methodology to other economic games (e.g., repeated prisoner's dilemma, trust games) is an important next step. Our framework and the identified behavioral archetypes provide a strong basis for such future investigations.

*   **Future Directions for Designing/Improving Cooperative LLMs (R-WHGr, R-gBFf):**

    *   Based on our findings, promising directions include: pro-social fine-tuning (explicitly training on datasets or with objectives rewarding collective well-being), reinforcement learning with group welfare objectives and exploring the impact of structured communication.
Detailed responses to each reviewer's unique points, and further elaboration on the above, are provided below.

**References**

[1] Gurerk, O., Irlenbusch, B., & Rockenbach, B. (2006). The Competitive Advantage of Sanctioning Institutions. *Science*.

---

### Decision · Program_Chairs · 2025-07-08

**Decision:**

Accept

**Comment:**

The three reviewers converge on the view that this paper tackles a timely and important question—how LLMs balance self-interest and collective well-being in cooperation—and that the experimental design is careful and methodologically sound. All reviewers highlight the observed cooperation patterns as a noteworthy and insightful finding for multi-agent research and better social outcomes.

The weaknesses pointed out by the reviewers are mitigated by the authors in the rebuttal process as follows.
- Robustness of the methods: The authors conducted targeted experiments altering key game parameters to evaluate the robustness of the methods.
- The scope of the paper: The authors pointed out that using PGG allows for direct comparison with robust human behavioral data from Gurerk et al. and subsequent large-scale replications.
- Synthetic Prompting: The authors' findings from the prompt test suggest that the core behavioral patterns largely persisted for both Llama-3.3-70B (maintaining high cooperation) and o1-mini, indicating that their main conclusions are not solely an artifact of the initial technical phrasing.


To sum up, all reviewers recommend the "accept" decision. I share their enthusiasm. Remaining limitations (e.g., extending this methodology to other economic games, comprehensive sensitivity analysis across a wider range of parameters and prompt styles, etc.) are typical for a first systematic study. I recommend that the authors provide a clearer summary and clarification about the limitations and future work in the revised version of this paper.